# Red and Processed Meat Consumption in Poland

**DOI:** 10.3390/foods11203283

**Published:** 2022-10-20

**Authors:** Katarzyna Stoś, Ewa Rychlik, Agnieszka Woźniak, Maciej Ołtarzewski

**Affiliations:** National Institute of Public Health NIH—National Research Institute, 00-791 Warsaw, Poland

**Keywords:** red meat, processed meat, amount and frequency of consumption, households, adults

## Abstract

The aim of the study was to assess the quantity and frequency of meat consumption, especially of red and processed meat, in Poland. The amount of meat consumed was assessed using data from household budget surveys undertaken in 2000, 2010, and 2020. The frequency of consumption was assessed using Food Propensity Questionnaire data from 1831 adults in 2019–2020. Poles consumed 1.35 kg of unprocessed red meat and 1.96 kg of total processed meat per person per month in 2020. The consumption of red meat was lower than in the two previous decades; the consumption of processed meat fluctuated. Pork was the most commonly consumed red meat: 40% of adults consumed pork 2–3 times a week. Beef and other unprocessed red meat were most often consumed less than once a month (29.1%). Processed meat was often consumed: 37.8% of adults ate cold cuts, and 34.9% ate sausages and bacon 2–3 times a week. The consumption of red and processed meat in Poland was high and frequent. In particular, the consumption of processed meat exceeded the recommendations and might increase the risk of chronic diseases. It is necessary to implement activities aimed at reducing the consumption of red and processed meat in Poland.

## 1. Introduction

Meat is a valuable source of many nutrients. It contains high-quality protein; minerals, especially iron and zinc; and several vitamins, such as vitamins A, D, B_12_, thiamine, and niacin. It can also provide the body with unfavourable saturated fatty acids or cholesterol. In addition, processed meat often contains high amounts of sodium [1,2].

The negative health effects of meat are reported mainly in the case of red and processed meat. High consumption of these types of meat may increase the risk of cardiovascular disease, stroke, type 2 diabetes, and metabolic syndrome [3,4,5,6,7]. Stronger evidence of increased risk of these diseases was found for processed meat, whereas the relationship between unprocessed red meat consumption and morbidity was not always confirmed [8,9]. High red and processed meat intake may also contribute to the incidence of certain cancers [10,11]. The International Agency for Research on Cancer (IARC) [12] classified red meat as probably carcinogenic to humans (Group 2A), and processed meat as carcinogenic to humans (Group 1). Positive associations were reported between red meat and colorectal, pancreatic and prostate cancer, and between processed meat and colorectal and stomach cancer. It is assumed that the unfavourable effects of red and processed meat result from the fat content, including saturated fatty acids, haem iron, heat-generated carcinogens, e.g., heterocyclic aromatic amines and, in the case of processed meat, also from the addition of salt and other preservatives [3,12].

Many experts recommend reduced red and processed meat consumption. This applies, inter alia, to the World Cancer Research Fund International/American Institute of Cancer Research (WCRF/AICR) and food-based dietary guidelines in many countries [13,14].

In the Farm to Fork Strategy [15], the European Commission highlights the need to modify the diet not only to reduce the risk of life-threatening diseases, but also to limit the impact of the food system on the environment. According to the Lancet Commission (EAT), the eaten and produced food, and how it was produced, will determine the health of people and planet. Many changes are required to prevent both reduced life expectancy and environmental degradation [16]. A modified diet should, above all, contain less red meat and processed meat, and more vegetables and fruit. Meat produces more emissions per unit of energy compared with plant-based food [17]. In terms of global emissions of carbon dioxide, animal-based food production accounts for 57%, whilst plant-based foods accounts for 29% [18]. Finding innovative ways of growing and producing food are important for sustainability transformations [19]. For some types of meat, the rearing of mammals usually leads to more emissions than that of poultry [17]. Moreover, livestock farming also accounts for a significant proportion of water consumption, which is largely used to irrigate feed crops [20]. About one-third of the world’s water consumption is used to produce animal products. Animal agriculture is responsible for 20 to 33% of all fresh water consumption in the world [21,22]. Daily water requirements for pigs, depending on age, range between 3 and 12 L/day and, for cattle, they range between 25 and 160 L/day [23].

For the study aims, it is important to define the terms “red meat” and “processed meat”. According to the IARC [12], the term red meat refers to unprocessed, fresh, or frozen meat derived from the muscle tissue of mammals, which is usually eaten cooked. This includes pork, beef, veal, mutton, lamb, goat, and horse meat. Processed meat means meat that has undergone the following processes: salting, smoking, curing, fermentation, or other processes improving preservation or flavour. According to the WCRF/AICR [13], red meat means all types of muscle meat from a mammal, including pork, beef, veal, mutton lamb, goat, and horse. Processed meat is considered to be transformed through salting, curing, fermentation, smoking, or other processes to improve preservation or enhance flavour, for example, ham, bacon, and sausage.

The consumption of meat is determined by sociodemographic factors. One of the most important factor is the sex of a consumer. Women reduce their meat consumption much more than men. Locality also has an influence on meat consumption. Residents of small cities and villages choose to eat meat more often than the inhabitants of large cities [24]. Consumers with higher levels of education and income are more easily able to adapt their dietary habits towards reducing meat in their diets or excluding it altogether. [25]. However, the influence of income on meat consumption choice is not obvious [26].

The aim of the study was to assess the current consumption of meat, especially red and processed meat, in Poland and the trends in this regard in the last two decades, based on a household budget survey. The aim of this study was also to analyse the frequency of selected meat and meat product consumption associated with sociodemographic factors in a representative sample of adult inhabitants in Poland.

## 2. Materials and Methods

### 2.1. Household Budget Survey Data Characteristics

#### 2.1.1. Household Budget Survey Selection and Sample

Meat consumption assessment in 2000, 2010, and 2020 was based on a household budget survey conducted throughout the country by Statistics Poland. The survey included a randomly selected sample of Polish households: 36,163 households in 2000; 37,412 in 2010; and 33,529 in 2020. The random sampling of the Polish household samples was based on the rules of Statistics Poland. The sampling scheme first included large cities (more than 100 thousand inhabitants), followed by the remaining towns and villages. The first-stage sampling frame was based on the records of statistical regions designed for the National Census, including dwelling specifications. The second-stage sampling frame was based on the registers of dwellings in the selected primary sampling units. Each month, different households participated in the survey [27].

Our study included data for all households, households depending on their locality (urban and rural), and socio-economic groups (employees, farmers, self-employed, retirees, and pensioners).

#### 2.1.2. Household Budget Survey Methodology Aspects

The Household Budget Survey, conducted annually by Statistics Poland, provides information on the living conditions of the population. In respect of food, it provides the information about the consumed amount expressed per one person per household. Each participating household receives a special diary, in which are registered, for a month, the food quantities purchased, obtained free, or derived from an individual farm, garden or business activity [27]. The methodology used does not include food quantities consumed in catering establishments, canteens, hospitals, nurseries, kindergartens, etc. [27].

#### 2.1.3. Meat Consumption Data

For the purposes of this study, the data collected by Statistics Poland were transformed to evaluate the consumption of various types of meat. The consumption of raw (unprocessed) and processed meat was calculated. Raw meat included total red meat and poultry. The Household Budget Survey does not make it possible to assess the consumption of individual types of red meat (e.g., pork, beef) separately. Processed meat included total red meat products and poultry products. The consumption of total red meat (processed and unprocessed) and total processed meat (red meat and poultry) were also analysed. Moreover, the consumption of offal and their products were calculated together.

### 2.2. Data from Nationwide Dietary Survey in Poland

#### 2.2.1. Study Population

These data were collected as part of the Nationwide Dietary Cross-Sectional Survey in Poland. This survey was conducted, from July 2019 to February 2020, on a representative sample of 2432 adolescents aged 10–17 years and adults (1214 males and 1218 females), according to the European Food Safety Authority (EFSA) guidance for the EU Menu methodology [28,29].

The study was approved by the Bioethics Committee at the Institute of Food and Nutrition in Warsaw, Poland (approval dated 4 June 2018). Participation in the study was voluntary, with each respondent providing written consent to participate.

The respondents were randomly selected from the PESEL register (the national register of inhabitants in Poland) [30] by the stratified sampling method, taking into account such demographic details as age (seven age cohorts), sex (female and male), as well as the size of the place of residence, and the territorial distribution within voivodships (16 voivodships, nine subdivisions of localities). The sample selection procedure consisted of the stratification of the Polish population and the random selection of individuals. Subjects who were hospitalised and/or were following enteral and parenteral nutrition because of their health conditions and subjects whose mental condition precluded the possibility of obtaining reliable information (e.g., neurodegenerative disease, drunkenness, state after taking drugs, and other stimulating substances), were excluded from the study. In order to prevent unforeseen circumstances or refusal to participate in the study and to ensure the planned number of people would be included in the study, the number of selected subjects was 10 times higher than that assumed. If a respondent refused to participate in the study, withdrew from the study, or met the exclusion criteria, another person from the group was selected. The interviewers contacted 4249 people and the response rate was 57%. The complete set of data was collected from 2432 adolescents and adults (1214 males and 1218 females); however, only adults (*n* = 1831, 913 men and 918 women) were included in the analysis of frequency of the consumption of selected meat and meat products.

#### 2.2.2. Data Collection

The socio-demographic data and the data of the frequency of consumption of selected types of meat and meat products were collected by the computer-assisted personal interviewing (CAPI) technique [31]. The socio-demographic questionnaire included questions about such characteristics as sex (male or female), age (years), educational level, as well as economic status. Educational level was classified as primary education/lower secondary education, vocational education, upper secondary education, post-secondary education, and higher education. The economic status was assessed according to the 5-item scale: very good, good, neither good nor bad, bad, very bad. Participants were also asked about their self-reported health status (both physical and mental health). Based on the questionnaire used by Statistics Poland in the European Health Interview Survey (EHIS) [32], the following question was used: “How is your health in general?” with five possible answers: “very good”, “good”, “neither good nor bad”, “bad” or “very bad”.

The Food Propensity Questionnaire (FPQ), part of the Nationwide Dietary Cross-Sectional Survey, was used to collect data about the frequency of consumption of selected meat and meat products. This questionnaire was based on a food frequency questionnaire from the Polish Academy of Science [33] and on the information from the project “Pilot study in the view of a Pan-European dietary survey-adolescents, adults and elderly (PANEU)” [34]. Respondents were asked about their frequency of consumption during the 12 months prior to the study, in terms of type of meat and meat products such as: “beef, veal, and mutton”, “pork”, “poultry”, “cold cuts, such as ham and loin”, and “sausages and bacon”. They could choose the following answers: “never”, “less than once a month”, “1–3 times a month”, “once a week”, “2–3 times a week”, “4–5 times a week”, “once a day”, or “several times a day”.

#### 2.2.3. Statistical Analysis

The results were statistically tested using the computer software PQStat 1.8.2. In the case of continuous data (age of respondents), in order to verify whether the distribution was normal, the Shapiro–Wilk test was used. The data distribution was not normal so the significance of difference was assessed using the Mann–Whitney U test for non-parametric data. The chi-square test was used for qualitative data. Relationships between socio-demographic characteristics and frequency of consumption of selected types of meat and meat products were examined using Spearman’s correlation. For all analyses, the significance level α = 0.05 was assumed.

## 3. Results

### 3.1. Results of Meat and Meat Product Consumption from Household Budget Survey

Meat and meat product consumption (Figure 1) in total households decreased by 7% between 2000 and 2020, and the decline was a little higher in urban households than in rural households. Considering the socioeconomic groups, the highest decrease in meat and meat product consumption was observed in the households of farmers.

According to the data from the Household Budget Survey, the consumption of unprocessed red meat in Poland, in 2020, was 1.35 kg/person/month estimated raw (Table 1). In the years 2000–2020, it decreased by 22%. The decline was noted both in urban and rural areas and in most of socio-economic groups, excluding pensioners. More unprocessed red meat was eaten in rural than in urban areas (by 20% in 2020). In 2020, the greatest amount of this type of meat was consumed in the households of retirees and pensioners, although, in 2000 and 2010, the highest consumption was observed in farmer households.

The consumption of red meat products (processed red meat), in 2020, was 1.75 kg/person/month. The decline occurred during the decade 2010–2020 and amounted to 11%. More red meat products (processed red meat) were consumed in rural than in urban areas (by 10% in 2020). Recently, most of these products were consumed in the households of retirees and pensioners.

Overall, the consumption of unprocessed and processed red meat, in 2020, was 3.10 kg/person/month. This was 15% lower than in 2000.

The consumption of poultry (estimated raw), in 2020, was 1.55 kg/person/month and was 16% higher than in 2000. The increase in consumption of this meat was recorded primarily in the years 2000–2010. In 2020, 15% more poultry meat compared with unprocessed red meat was consumed in Poland. Red meat accounted for 47% of the total unprocessed meat consumed.

The consumption of processed poultry products in 2020 was 0.21 kg/person/month and was 50% higher than in 2000. The consumption of poultry products increased in the first decade of the 21st century. It did not change in the second decade. In recent years, the consumption of processed poultry products in urban and rural areas was similar. Data for specific types of households indicated that, recently, members of pensioner households ate the most poultry products.

The consumption of processed meat, both red and poultry, in 2020, was 1.96 kg/person/month, of which 89% was red meat. During the first two decades of the 21st century, the consumption of processed meat fluctuated: in the first decade, it increased by 7%, and in the second decade, it decreased by 11%.

In 2020, offal and offal products were consumed in the amount of 0.23 kg/person per month. Compared with 2000, a decrease of 36% was noted. More of these products were consumed in rural than urban areas (by 10% in 2020). The highest consumption was observed in the households of retirees and pensioners.

### 3.2. Results of Nationwide Dietary Survey in Poland

#### 3.2.1. Subject Characteristics

A total of 90% of the 1831 subjects (89% of 812 men and 91% of 839 women) declared that they consumed meat and meat products during the 12 months prior to the study. Among respondents who declared meat consumption, 49.2% were men and 50.8% were women. The average age of subjects consuming meat was 51.9 ± 19.7 (18–96) years and it was the same for both men and women (*p* = 0.9912). Most respondents had an upper secondary education (41.7%). There were more women than men with a higher educational level (15.6 vs. 8.3%) and more men than women with a vocational education (39.5 vs. 29.5%), (*p* < 0.0001). Most subjects described their economic status as “neither good nor bad” (54.4%), followed by “good” (34.7%). In the case of health status, respondents most often chose the answer “good” (42.9%), followed by “neither good nor bad” (34.9%). There were no statistically significant differences between men and women in the case of these two parameters (Table 2).

#### 3.2.2. Frequency of Meat and Meat Product Consumption during the 12 Months Prior to the Study

The subjects consumed beef, veal, and mutton relatively rarely (Table 3). Most subjects declared that they consumed meat from this category less than once a month (29.1%). A similar percentage of subjects (27.6%) reported the consumption of meat from this category as 1–3 times a month. As many as 17.6% of respondents had never consumed beef, veal, or lamb during the 12 months prior to the study. The frequency of consumption of meat in this food category was similar in men and women.

Pork consumption was quite frequent during the 12 months prior to the study. In the total group, most subjects consumed pork 2–3 times a week (40%) or once a week (37.3%). Men were more likely to consume pork than women (*p* < 0.0001). Specifically, more men than women (46.1% vs. 34.1%) reported eating this type of meat 2–3 times a week, whereas women tended to eat pork once a week compared with men (42% and 32.5%, respectively).

Most subjects (59%) consumed poultry 2–3 times a week. There were also other frequent answers: “once a week” (22.1%) or “4–5 times a week” (12.2%). Only a few subjects had never eaten poultry during the 12 months prior to the study (0.4%). There were no significant differences in the frequency of consumption of poultry between men and women.

The consumption of cold cuts, such as ham and loin, by subjects was frequent. Most reported consumption of such meat products as 2–3 times a week (37.8%) or 4–5 times a week (23.7%). A number of subjects also reported eating cold cuts “once a week” (15.6%). The frequency of consumption of such meat products was similar in men and women.

Sausages and bacon were consumed most often 2–3 times a week (34.9% across the entire group) or once a week (25.4%). Men were more likely to consume these products than women (*p* < 0.0001). In particular, more men than women reported eating sausages and bacon 2–3 times a week (41.2% vs. 28.9%) and 4–5 times a week (14.6% vs. 8.4%), whereas more women than men consumed these meat products 1–3 times a month (19.3% vs. 10%). 

#### 3.2.3. Relationships between Socio-Demographic Characteristics, Health Status, and Frequency of Consumption of Selected Types of Meat and Meat Products

Across the entire group, a significant positive relationship was observed between the frequency of the consumption of beef, veal, and mutton, and education level, as well as economic status (Table 4). In men, such relationships were also noted; in addition, a relationship was found between their health status and the frequency of consumption of this type of meat. In women, only the relationship between economic situation and the frequency of consumption of beef, veal, and mutton was observed.

The frequency of pork consumption was negatively related to sex and education level across the group: age in men and education level in women.

In the case of poultry, a significant negative relationship was observed between the frequency of consumption of this type of meat and age across the entire group, i.e., for both men and women. In the entire group and in women, a positive relationship was found between the frequency of poultry consumption and economic status.

The frequency of consumption of cold cuts, such as ham and loin, was negatively related to sex and age and positively related to economic status and health status across the group, both in men and women.

A negative correlation was observed between the frequency of consumption of sausages and bacon and sex across the whole group, and between the frequency of consumption of these type of meat products and education level across the whole group, in both men and women.

## 4. Discussion

Dietary recommendations often relate to the consumption of meat, including red and processed meat. The food-based dietary guidelines (FBDGs) of most European countries recommend limiting meat consumption to around 300–600 g per week and not eating meat every day. Some FBDGs indicate that especially red and processed meat intake should be reduced. The maximum amount of these types of meat is usually given as up to 500 g/week (e.g., Finland, Sweden), or less than twice weekly (e.g., Malta) [14]. The recommendations of many countries mention fish, eggs, legumes, mycoprotein-based foods, and seitan as meat substitutes [2,14].

In the Polish FBDG, special attention is paid to limiting the consumption of red and processed meat to 500 g per week [35]. It is recommended that eating meat be avoided at least 1 day a week, poultry be chosen, and meat be replaced with fish, eggs, and plant products such as legumes and nuts. A meta-analysis by Guasch-Ferré et al. indicates that replacing red meat with high-quality plant protein sources, but not with fish or low-quality carbohydrates, provides more favourable changes in blood lipids and lipoproteins. Red meat resulted in greater decreases in total cholesterol than fish [5]. In the case of fish, the method of their cooking is very important. Eating fish can reduce the incidence of cardiovascular disease but eating fried fish can increase the risk of cardiovascular events [36,37]. In Poland, it is particularly important to include fish in food-based dietary guidelines due to very low consumption in this country [27,38].

The World Cancer Research Fund/American Institute of Cancer Research (WCRF/AICR) recommends limiting red meat consumption to no more than about three portions per week [13]. According to their estimates, three portions are equivalent to about 350–500 g cooked weight and 500 g of cooked red meat is equivalent to about 700 to 750 g of raw meat. It is also recommended to consume very little, if any, processed meat [13]. The European Food Safety Authority (EFSA) analysed the most recent systematic reviews of prospective cohort studies and estimated that the risk of chronic diseases, especially cancer, increases at a consumption of 100–120 g/day of unprocessed red meat and 50 g/day of processed meat [2]. These amounts are above most current recommendations in European national FBDGs.

Data from the Household Budget Survey presented in this study indicate that the consumption of unprocessed red meat in Poland in 2020 was 311 g/person/week, whilst the consumption of processed meat, including red, was 451 g/person/week. Although, in 2020, these amounts were lower than in 2000, the consumption of red and processed meat in Poland still exceeds the national recommendations [35]. Particularly noteworthy is the high proportion of processed meat. In relation to the WCRF/AICR recommendations [13], the consumption of red and especially processed meat in Polish households should be assessed as too high. Taking into account the EFSA conclusions, consumption of unprocessed red meat did not increase the risk of chronic disease [2]. The consumption of processed meat in an amount of 1.3 times more than 50 g/day could increase the risk of non-communicable diseases in Poland.

It should be emphasised that the overall changes in the quantity and structure of meat consumption in Poland in the last two decades were favourable. In 2000–2010, there was an increase in poultry consumption in place of unprocessed red meat. In the second decade, the consumption of unprocessed red meat continued to increase, while the consumption of poultry did not change. In recent years, Poles consumed more poultry than unprocessed red meat. This could be due to the fact that, in the years 2010–2020, the price of meat increased in Poland, but poultry prices to a lesser extent than pork and beef prices [39,40,41]. Following a slight increase in processed meat consumption between 2000 and 2010, there was a decline in the following decade, especially for red meat, which could result from price increases and, possibly, from a growing interest in healthy eating. In 2009, Poland developed revised dietary guidelines, which recommended eating meat in moderation, including limiting the consumption of red meat [42].

There were different trends in meat consumption in recent years in various countries. The comparison of meat consumption in 35 countries in 2000–2019, based on the OECD-FAO Agricultural Outlook database showed that global meat consumption increased between 2000 and 2019, from 29.5 kg to 34.0 kg per capita per year [43]. Above all, the consumption of pork and poultry increased and more mutton and lamb were eaten, while the consumption of beef and veal decreased. The most substantial increase for total meat was observed in countries with consumption below the world average in 2000, such as Russia, Vietnam, and Peru. Six countries recorded a decline in meat consumption in 2000–2019, most notably New Zealand, from 86.7 kg/capita to 75.2 kg/capita, and Paraguay, from 53.5 kg/capita to 39.5 kg/capita, per year. Poland was not included in this analysis.

A survey conducted in the UK, in individuals aged 1.5–96 years, using unweighted food diaries, showed that between 2008–2009 and 2018–2019, average meat consumption decreased by approximately 17.4 g per person per day [44]. The consumption of red and processed meat decreased by 13.7 g and 7.0 g, respectively, while the consumption of white meat (including poultry and fish) increased by 3.2 g.

Among US adults in 1999–2016, only a reduction in unprocessed red meat consumption was observed, from 340 g/week in 1999–2000 to 284 g/week in 2015–2016 [45]. During these 18 years, the consumption of processed meat remained unchanged and, in 2015/2016, it was 187 g/week. Poultry consumption increased from 256 g/week in 1999–2000 to 303 g/week in 2015–2016.

A review of data from nationally representative nutrition surveys from 2003 to 2014 in ten European countries showed that meat consumption varied across countries [46]. Total meat consumption in adults ranged from 75 to 93 g/day in Sweden to 191–211 g/day in Finland. Data on red and processed meat consumption were available only for the Netherlands, Ireland, and the UK. Mean intakes of red and processed meat were 71–93 g/day in younger adults and 63–85 g/day in older adults. Only in the UK, consumption of these types of meats was below the upper limits recommended by the WCRF/AICR, while in the Netherlands and Ireland, consumption exceeded this recommendation.

Our survey on food frequency showed that pork was the most commonly consumed unprocessed red meat in Poland. Other types of red meat were eaten less frequently. Unprocessed pork was most often eaten 2–3 times a week, which was consistent with the dietary recommendations. Poultry was eaten more often. Moreover, processed meat products were eaten very often; most of the respondents consumed them 2–3 times a week, and a large group even more frequently.

The study of the frequency of consumption of selected products carried out in 2019 in one region in Poland—the Subcarpathian voivodeship—also showed that poultry was consumed more often than red meat [47]. This could also be explained as a result of price increases for red meat and changes in eating habits. Cold cuts and other processed meat were often eaten.

A comparison of studies conducted in the United States, Canada, and Mexico in 2013–2016 showed that the consumption of red and processed meat was high in all these countries [48]. The median intake of unprocessed red meat estimated using cooked weight in Canada was 79.0 g/day; in the US, it was 72.3; and in Mexico, it was 62.0 g/day. The consumption of processed meat was slightly lower than that of unprocessed red meat: 41.8 g/day in Canada, 44.5 in the US, and 40.0 in Mexico.

In Poland, we observed a different situation: the consumption of processed meat exceeded the consumption of unprocessed red meat. It should be noted that the data from the Household Budget Survey refer to raw, not to cooked meat as eaten. Poles often eat sandwiches, especially for breakfast; a frequent addition to sandwiches are cold cuts and sausages. This can cause a higher consumption of processed meat [38,49]. In British studies from 2008 to 2019, the consumption of red and processed meat was similar; in 2018/2019 it was 34.7 and 36.1 g/person/day, respectively [44].

The Household Budget Survey showed that there was a higher consumption of meat, including red and processed meat, in rural areas compared with that in urban areas. In Poland, rural residents traditionally consume greater amounts of most food products compared with urban residents [50]. This is justified by the type of work undertaken in the countryside, which requires greater energy expenditure, and by easy access to products from their own farms. Moreover, it is quite common in the countryside to allocate some food products to animals, which in the Household Budget Survey may be shown as having been consumed by members of the household. Despite changes in the Polish diet in recent years, differences in consumption in urban and rural areas still persist. The aforementioned study in the Subcarpathian voivodeship showed that the village inhabitants more often mainly consumed staple foods, such as cereals and potatoes. In the case of meat, there were significant differences for cold cuts, which were eaten more often in rural than in urban areas [47].

In some countries, meat consumption also varies depending on the place of residence. Results for the German National Nutrition Survey II from 2005 to 2007 indicated that high meat consumption was more prevalent among persons living in small cities and rural areas in comparison with those living in large cities [24].

The households of retirees and pensioners were characterised by the highest consumption of meat. These households had the highest consumption of many food products per person, which resulted from their number and age structure. In 2020, people aged 14 or younger accounted for 13.4% of household members included in the budget survey, while in the households of retirees and pensioners it was 1.5% and 6.3%, respectively [27]. The average number of persons in a household, taking into account all households, was 2.61, while in the households of retirees and pensioners, it was 1.81 and 1.64, respectively. The number of persons in a household affects the level and structure of expenses. Expenditure on food and non-alcoholic beverages per capita is the highest in small households, which can be associated with the economy scale effect related to savings in the joint preparation of meals [51].

The data on food frequency described in this manuscript showed that gender influenced the consumption of most of the analysed types of meat. Men ate pork, cold cuts, sausages, and bacon more often than women. The review of data from ten European countries showed that mean intakes of total meat across Europe were higher in men (84–218 g/day) than in women (64–163 g/day) [46]. The aforementioned national survey in Germany indicated a much higher meat consumption by men (155 g/day) than by women (87 g/day) [24]. However, according to UK data from the National Diet and Nutrition Survey rolling programme (2008–2009 to 2018–2019), there was no significant difference in meat intake between men and women from the UK when the intake was expressed as a percentage of food energy [44]. The relationship between sex and meat consumption was also assessed by the authors of a nationally representative cross-sectional survey comparison of dietary recalls from Canada, Mexico, and the United States. In the US and Canada, unprocessed red and processed meat intake was higher for men than for women among consumers. No such relationship was found in Mexico [48].

Our survey found a relationship between the frequency of meat consumption and age. The elderly consumed poultry and cold cuts less frequently than younger people and, in men, the frequency of consumption of unprocessed pork decreased with age. Mean consumption of total meat in most of the ten European countries reviewed was higher in younger adults aged 18–69 years (93–233 g/day) than in older adults aged over 65 years (75–191 g/day) [46]. The trend of reducing meat consumption with increasing age was observed in Germany [24]. In British studies, older adults consumed less meat, expressed as a percentage of food energy, compared with younger adults [44]. In Mexico, the consumption of total meat in the elderly was lower than in the younger age groups. In Canada and the US, elderly people consumed less meat; however, the differences were not meaningful [48].

Meat consumption can be influenced by level of education. In our survey, as the education level increased, the frequency of consumption of sausage and bacon, and in women, also of pork, decreased. On the other hand, the frequency of beef, veal, and mutton consumption increased with the growing education level among men. In Poland, these types of meat are among the more expensive. A certain correlation between the level of education and meat consumption was demonstrated in studies conducted in the Subcarpathian voivodeship, Poland [47]. Subjects with a lower level of education were more likely to consume processed meat, with the exception of cold cuts, than those with a higher level of education. However, there was no correlation between level of education and the consumption of red meat, poultry, and cold cuts.

In other countries, the level of education also influenced meat consumption. A Slovenian study showed that the education level had an impact on the frequency of meat consumption and sustainability attitudes; a higher educational level was an independent predictor of lower meat consumption [52]. In Germany, consumers characterised by a higher educational level more often reduced meat consumption than people with a lower educational level [24]. In the United States and Canada, the level of education was a factor influencing unprocessed red and processed meat consumption [48]. Subjects with a lower level of education were more likely to consume these types of meat compared with subjects with a higher level of education. However, there were no associations between educational level and meat consumption in Mexico.

Economic status had an impact mainly on the consumption of more expensive products: beef, veal, and mutton, as well as cold cuts. These were more often eaten by subjects with better financial standing. Income affects meat consumption mainly in Mexico: subjects with a high income consumed more unprocessed red and processed meat than those with low and middle incomes [48]. In Canada, high income was associated with a higher consumption of processed meat. In the US, the consumption of unprocessed red meat was higher for those with a middle income than for those with a high income [48]. An analysis of meat consumption in 35 countries showed that changes in consumption in some countries were positively related to changes in GDP, while in others they were not related to GDP [43]. The authors of this paper argue that GDP growth leads, to some extent, to an increase in meat consumption; however, with a GDP of around USD 40,000 per capita, the increase in economic well-being in the country does not lead to growth in meat consumption [43].

The impact of socio-demographic factors on food choice is quite complex. A consumer study by Einhorn found that younger and well-educated respondents primarily perceive their diet as something with which to experiment. Mostly highly educated and financially well-off people want to try new foods and have advanced knowledge of nutrition. When time or money is scarce, respondents show little interest in food planning, purchasing, and preparation [25].

The inclusion of data from the Household Budget Survey and data on the frequency of food consumption in this manuscript allowed for a more complete characterisation of the red and processed meat intake in Poland. Consumption and changes in the last two decades were analysed, taking into account the place of residence and the type of household. The frequency of consumption of various types of meat among adults and the impact of various socio-economic factors were also assessed.

This study also had several limitations. Data from the Household Budget Survey show the average consumption per capita, without considering age, gender, or other characteristics which are very valuable especially for various comparisons, and to a lesser extent reflect actual consumption. Moreover, the categories of meat were slightly different in both data sources. The Household Budget Survey provided information on total raw red meat consumption, disregarding the source of origin. The food frequency data, on the other hand, provided information on the consumption of processed meat products according to categories (cold cuts, sausages, and bacon) without specifying whether they were red meat or poultry.

## 5. Conclusions

Although in recent years, in Poland, there was a noted decline in the consumption of red meat, both unprocessed and processed, the consumption of this type of food is still high, especially the amount of processed meat in the diet, which exceeds the recommendations. A high consumption of processed meat may increase the risk of chronic diseases. More meat and processed products were eaten in rural than in urban areas. The highest consumption was noted in the households of retirees and pensioners. Among the different types of meat, poultry was often consumed. This seems positive because this type of meat is recommended as a replacement for red meat. Unfortunately, the frequent consumption of pork, especially among men, and processed meat (cold cuts, such as ham and loin, and for men, sausages and bacon) was also observed. Depending on the type of meat and meat products, relationships were observed between the frequency of their consumption and socio-demographic characteristics, such as sex, age, education level, economic status, and health status. In Poland, it is very important to follow up activities aimed at reducing the amount and frequency of consumption of mainly various types of processed meat, as well as unprocessed red meat. There is a need to educate the public in the field of healthy eating, including the health effects of consuming too much red meat and processed meats. Education about sustainable development and the environmental impact of too large a meat production industry is also required. Educational activities should be directed, in particular, at the population groups most exposed to the negative health effects caused by the excessive consumption of red and processed meat.

## Figures and Tables

**Figure 1 foods-11-03283-f001:**
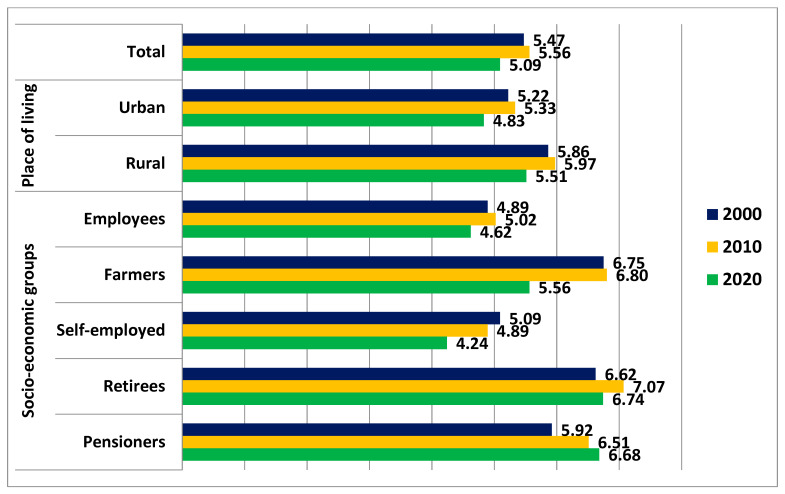
Total meat and meat product consumption in Polish households in 2000, 2010, and 2020 in terms of place of living and socio-economic group (kg/person/month).

**Table 1 foods-11-03283-t001:** The consumption of meat and meat products in Polish households in 2000, 2010, and 2020 with in terms of place of living and socio-economic groups (kg/person/month).

Type of Meat	Year	Total	Place of Living		Socioeconomic Groups				
			Urban	Rural	Employees	Farmers	Self- Employed	Retirees	Pensioners
Raw red meat	2000	1.73	1.62	1.90	1.53	2.49	1.70	2.02	1.75
	2010	1.57	1.47	1.74	1.40	2.36	1.45	1.95	1.64
	2020	1.35	1.25	1.50	1.19	1.61	1.13	1.85	1.72
Processed red meat products	2000	1.90	1.85	1.96	1.80	2.05	1.77	2.19	1.99
	2010	1.97	1.92	2.04	1.82	2.17	1.71	2.42	2.22
	2020	1.75	1.68	1.84	1.61	1.82	1.45	2.25	2.23
Raw red meat and processed red meat products	2000	3.63	3.47	3.86	3.33	4.54	3.47	4.21	3.74
	2010	3.54	3.39	3.78	3.22	4.53	3.16	4.37	3.86
	2020	3.10	2.93	3.34	2.80	3.43	2.58	4.10	3.95
Raw poultry meat	2000	1.34	1.25	1.49	1.12	1.70	1.21	1.71	1.56
	2010	1.52	1.46	1.62	1.36	1.67	1.33	2.00	1.98
	2020	1.55	1.47	1.70	1.42	1.67	1.34	2.02	2.09
Poultry processed meat products	2000	0.14	0.16	0.12	0.15	0.08	0.15	0.19	0.14
	2010	0.22	0.22	0.23	0.21	0.20	0.20	0.25	0.26
	2020	0.21	0.21	0.21	0.21	0.19	0.18	0.22	0.27
Red and poultry processed meat products	2000	2.04	2.01	2.08	1.95	2.13	1.92	2.38	2.13
	2010	2.19	2.14	2.27	2.03	2.37	1.91	2.67	2.48
	2020	1.96	1.89	2.05	1.82	2.01	1.63	2.47	2.50
Offal and processed offal products	2000	0.36	0.34	0.39	0.29	0.43	0.26	0.51	0.48
	2010	0.28	0.26	0.34	0.23	0.40	0.20	0.45	0.41
	2020	0.23	0.22	0.26	0.19	0.27	0.14	0.40	0.37

Source: Author calculations based on Statistics Poland data.

**Table 2 foods-11-03283-t002:** Characteristic of subjects.

Parameter	Total (*n* = 1651)		Men (*n* = 812)		Women (*n* = 839)		M vs. W
	*n*	%	*n*	%	*n*	%	*p*
Education level:							
Primary education/lower secondary education	169	10.2	82	10.1	87	10.4	<0.0001
Vocational education	552	33.4	321	39.5	231	27.5	
Upper secondary education	688	41.7	333	41.0	355	42.3	
Post-secondary education	44	2.7	9	1.1	35	4.2	
Higher education	198	12.0	67	8.3	131	15.6	
Economic status:							
Very bad	10	0.6	5	0.6	5	0.6	0.0600
Bad	102	6.2	37	4.6	65	7.8	
Neither good nor bad	898	54.4	462	56.9	436	52.0	
Good	572	34.7	276	34.0	296	35.3	
Very good	69	4.2	67	8.3	37	4.4	
Health status:							
Very bad	6	0.4	3	0.4	3	0.4	0.9712
Bad	63	3.8	33	4,1	30	3.6	
Neither good nor bad	576	34.9	287	35.4	289	34.5	
Good	709	42.9	346	42.6	363	43.3	
Very good	297	18.0	143	17.6	154	18.4	

**Table 3 foods-11-03283-t003:** Frequency of consumption of selected type of meat and meat products.

Type of Meat	Sex	Frequency of Consumption																M vs. W
		Never		Less than Once a Month		1–3 Times a Month		Once a Week		2–3 Times a Week		4–5 Times a Week		Once a Day		Several Times a Day		* *p*
		*n*	%	*n*	%	*n*	%	*n*	%	*n*	%	*n*	%	*n*	%	*n*	%	
Beef, veal, and mutton	Total	289	17.6	477	29.1	453	27.6	297	18.1	97	5.9	23	1.4	4	0.2	2	0.1	
	Men	140	17.3	220	27.2	229	28.3	150	18.5	55	6.8	13	1.6	1	0.1	1	0.1	0.5374
	Women	149	17.9	257	30.9	224	26.9	147	17.6	42	5.0	10	1.2	3	0.4	1	0.1	
Pork	Total	32	1.9	53	3.2	187	11.4	613	37.3	657	40.0	87	5.3	13	0.8	1	0.1	
	Men	12	1.5	20	2.5	82	10.1	263	32.5	373	46.0	52	6.4	7	0.9	1	0.1	<0.0001
	Women	20	2.4	33	4.0	105	12.6	350	42.0	284	34.1	35	4.2	6	0.7	0	0.0	
Poultry	Total	6	0.4	12	0.7	61	3.7	365	22.1	973	59.0	201	12.2	29	1.8	2	0.1	
	Men	2	0.2	6	0.7	33	4.1	166	20.5	483	59.6	105	12.9	14	1.7	2	0.2	0.5273
	Women	4	0.5	6	0.7	28	3.3	199	23.7	490	58.5	96	11.5	15	1.8	0	0.0	
Cold cuts, such as: ham, loin	Total	44	2.7	41	2.5	75	4.6	257	15.6	623	37.8	391	23.7	143	8.7	73	4.4	
	Men	18	2.2	21	2.6	34	4.2	123	15.2	294	36.3	206	25.4	79	9.7	36	4.4	0.4387
	Women	26	3.1	20	2.4	41	4.9	134	16.0	329	39.4	185	22.1	64	7.7	37	4.4	
Sausages and bacon	Total	46	2.8	110	6.7	242	14.7	418	25.4	575	34.9	188	11.4	60	3.6	7	0.4	
	Men	10	1.2	38	4.7	81	10.0	186	22.9	334	41.2	118	14.6	40	4.9	4	0.5	<0.0001
	Women	36	4.3	72	8.6	161	19.3	232	27.8	241	28.9	70	8.4	20	2.4	3	0.4	

M vs. W—men versus women. * chi-square test.

**Table 4 foods-11-03283-t004:** Relationships between socio-demographic characteristics and the frequency of consumption of selected types of meat and meat products.

Type of Meat	Parameter	Sex *		Age		Education Level		Economic Status		Health Status	
	Sex	Rs	*p*	rs	*p*	rs	*p*	rs	*p*	rs	*p*
Beef, veal, and mutton	Total	−0.0395	0.1092	−0.0146	0.5544	0.0529	0.0320	0.0799	0.0012	0.0430	0.0814
	Men			−0.0116	0.7413	0.0792	0.0243	0.0922	0.0087	0.0793	0.0242
	Women			−0.0166	0.6318	0.0430	0.2154	0.0691	0.0462	0.0092	0.7906
Pork	Total	−0.1391	<0.0001	−0.0195	0.4295	−0.0807	0.0011	−0.0005	0.9855	−0.0090	0.7156
	Men			−0.0967	0.0059	−0.0561	0.1105	−0.0158	0.6526	0.0318	0.3654
	Women			0.0563	0.1047	−0.0697	0.0443	0.0134	0.6991	−0.0443	0.2012
Poultry	Total	−0.0327	0.1848	−0.1190	0.0000	0.0147	0.5514	0.1063	0.0000	0.0430	0.0809
	Men			−0.1446	0.0000	−0.0051	0.8846	0.0675	0.0546	0.0658	0.0612
	Women			−0.0946	0.0061	0.0365	0.2907	0.1414	0.0000	0.0233	0.5006
Cold cuts, such as ham, loin	Total	−0.0501	0.0419	−0.1053	0.0000	0.0308	0.2119	0.1274	<0.0001	0.1355	<0.0001
	Men			−0.1191	0.0007	0.0190	0.5891	0.1502	0.0000	0.1458	0.0000
	Women			−0.0923	0.0076	0.0542	0.0542	0.1060	0.0021	0.1264	0.0002
Sausages and bacon	Total	−0.2291	<0.0001	0.0045	0.8538	−0.1138	0.0000	0.0018	0.9426	0.0009	0.9697
	Men			−0.0492	0.1619	−0.0798	0.0231	−0.0437	0.2139	0.0210	0.5494
	Women			0.0618	0.0744	−0.0903	0.0090	0.0409	0.2381	−0.0158	0.6477

* Range for men: 1, range for women: 2. rs—Spearman’s rho correlation coefficient.

## Data Availability

Data are available on reasonable request. The dataset used to conduct the analyses is available from the corresponding author on reasonable request.

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
