# Peer review of "Red and Processed Meat Consumption in Poland"

_foods, 2022, doi:10.3390/foods11203283_

Round 1
Reviewer 1 Report
This is an important area to study for the reasons outlined and this can indeed make an important contribution to work in the field. Overall a very interesting and promising article.
This article seems it is not only about red and processed meat consumption in Poland but also around the consequences from this consumption and possible future interventions, so in the abstract part I recommend adding this as aim (see Abstract L8-9)
Great example with Farm to Fork strategy L48. It should be worth looking also at EAT Lancet Planetary diet Willett at al., 2019 as well will give another supportive and interesting perspective.
What is the exact emissions footprint meat produces? L51 ”Meat produces more emissions per unit of energy compared to plant-based food” Perhaps check few sources:
Xu, et al., 2021, “Global greenhouse gas emissions from animal-based foods are twice those of plant-based foods” https://www.nature.com/articles/s43016-021-00358-x
Marinova, D., Bogueva, D. (2022) Food in a planetary emergency, Nature. https://link.springer.com/content/pdf/bfm:978-981-16-7707-6/1.pdf
Reference needed about different emissions per type of meat L52 “For types of meat, the rearing of mammals usually leads to more emissions than that of poultry.”
Same with the water L54-55. Provide references. It is not only for irrigation. Animal agriculture is responsible for up to one third of all fresh water consumption in the world today. High-producing milking cows can consume up to 200 L/day of water, while sheep can drink 40% more during summer than winter.
A little bit of a jump to suddenly announce the need to look for a definition. L56 “It is important to adopt the definition of red and processed meat”. More appropriate will be to tailor this need with the study on which this article is focused on, something in line: For the study aims etc...
L70 add another aim toward …. consequences from this consumption and possible future interventions. This could strengthen and improve the article.
Methodology
Make sure you clearly state that the study presented in this article is based on analysis of a secondary data from Statistics Poland from the start. I noted it is present in L111. Also, is this data publicly available? L133”The interviewers tried to contact 4249 people” Please
L77 How the random sampling was done? L79-82 not so clear.
L82 Which households? Perhaps Polish, which country areas, major cities.
How the Household budget survey was distributed? Is this part of the Statistics Poland national survey or your study own? It is not quite clear. L88-89 How the survey established knowledge of the following living conditions: “food consumption, housing conditions and equipment of the households with durables”.
L89-91 needs clarity.
If there is no concrete data of the food quantities in catering establishments L93-94, please give references to the usual quantities, what is the average serve?
L96 Why it is internationally underestimated?
L137- 138 “From among adults, persons who declared in the survey that they consumed meat were 137 selected (n=1,651, 812 men, 839 women).” Please rewrite for clarity.
L155-164 Polish Academy of Science survey, household budget survey, Statistics Poland, PANEU etc.? It’s a bit confusing on which exactly study data collections are based this article. Please make it clear.
Before the concrete results by year are presented is needed a sentence or two explaining the overall results across years like meat consumption of both types processed and unprocessed meat varied between years…… and etc. L183-186 Why do you think such observation was noted “unprocessed red meat was eaten in rural than in urban area”? Same with red meat L187-190. Are there any other events that predisposed such results? Why do you think these products were consumed in the household of retirees and pensioners (L190, L204 in relation to chicken)? What the data reveals? Are there any explanations for this trend? Some is explained L391-400 which is great.
L201 Why the poultry consumption has not changed in the last decade?
Table 1 needs to be presented also in a graph for the trend to be seen.
Discussion
L300 These references are great but how these are applicable to the results of the presented study? Please make your case, not just repeating the data shown in the survey results L323-325..
L332-333 What this leads to? Why it is good or bad?
L340 Why it is a decline in red meat? Any particular reasons? Some trends? Price increase, availability problems etc.?
L377-378 -Why poultry was more often consumed than red meat.
L386 “In Poland, we observed a different situation: the consumption of processed meat 386 exceeded the consumption of unprocessed red meat. It should be noted that the data 387 from the household budget survey refer to raw, not to cooked meat.” Is a very interesting finding. Why do you think is that?
Combine the single sentence paragraphs (starting L40 and L44) and L197-204; L209-212; L301; L389; L432 etc. as usually there are 3 to 8 sentences in a paragraph. Please check the whole text as there are some tendency.
L402 Are there any explanations for the “differences for cold cuts, which were eaten more often in rural than in urban areas”?
Please give arguments for presenting the studies in the US, Canada, British and Mexico L417-419 and L432. Is this in comparison with your Polish study? It is brought a bit abrupt and without any linking explanation.
Full stop in missing after “sources” L92
L238 The subheadings 3.2.2. and 3.2.4. can’t be distinguished from the text. Make it italic or bold depending on the journal’s requirements.
L281 Full stop is not needed at the end of the subheading.
Conclusion must be a bit more conclusive as the moment seems like it is presenting a summary in bullet points without actually including them which is not necessary. Although the major points were summarized, the conclusion did not fully utilize the chance the authors to have their last word on the subject. Try to make it more cohesive.
The authors should be complimented on taking on this important area of research.
Author Response
AUTHORS’ RESPONSE TO THE COMMENTS FROM THE REVIEWER#1
Thank you for all the comments. We have carefully analysed all valuable comments and suggestions that allow the Authors to increase the scientific soundness of our paper. The authors modified the manuscript to improve the quality of the paper. Our responses are listed below.
COMMENT#1
This article seems it is not only about red and processed meat consumption in Poland but also around the consequences from this consumption and possible future interventions, so in the abstract part I recommend adding this as aim (see Abstract L8-9)
RESPONSE#1
Thank you very much for the comment. The purpose of the study was to evaluate consumption using the budget method and to assess the frequency of consumption on the basis of the food propensity questionnaire method. We would like to underline that the study did not investigate the effect of red and processed meat consumption on health.
COMMENT#2
Great example with Farm to Fork strategy L48. It should be worth looking also at EAT Lancet Planetary diet Willett at al., 2019 as well will give another supportive and interesting perspective.
RESPONSE#2
Thank you very much for the suggestion. The proposed reference was added:
“According to the Lancet Commission (EAT) the eaten and produced food and how it was produced will determine the health of people and planet. A lot of changes should be done to avoid both reduced life expectancy and environmental degradation [16].”
COMMENT#3
What is the exact emissions footprint meat produces? L51 ”Meat produces more emissions per unit of energy compared to plant-based food” Perhaps check few sources:
Xu, et al., 2021, “Global greenhouse gas emissions from animal-based foods are twice those of plant-based foods” https://www.nature.com/articles/s43016-021-00358-x
Marinova, D., Bogueva, D. (2022) Food in a planetary emergency, Nature. https://link.springer.com/content/pdf/bfm:978-981-16-7707-6/1.pdf
RESPONSE#3
Thank you very much for the suggestion. The proposed reference was added:
“In the global emissions of carbon dioxide the share of animal based food production is 57% whilst the plant based foods covers 29% [18]. Finding the innovative ways of growing and producing food are the important part of sustainability transformations [19].”
COMMENT#4
Reference needed about different emissions per type of meat L52 “For types of meat, the rearing of mammals usually leads to more emissions than that of poultry.”
RESPONSE#4
Thank you very much for the remark. The reference was included.
“For types of meat, the rearing of mammals usually leads to more emissions than that of poultry [17].”
COMMENT#5
Same with the water L54-55. Provide references. It is not only for irrigation. Animal agriculture is responsible for up to one third of all fresh water consumption in the world today. High-producing milking cows can consume up to 200 L/day of water, while sheep can drink 40% more during summer than winter.
RESPONSE#5
Thank you very much for the suggestion. The references were found and following text was added.
“About one-third of the world’s water consumption is for producing animal products. Animal agriculture is responsible for 20 to 33 percent of all fresh water consumption in the world [21,22]. Daily water requirements of pigs depending on age is ranged between 3–12L/day and for cattle is ranged between 25-160 l/day [23]
COMMENT#6
A little bit of a jump to suddenly announce the need to look for a definition. L56 “It is important to adopt the definition of red and processed meat”. More appropriate will be to tailor this need with the study on which this article is focused on, something in line: For the study aims etc...
RESPONSE#6
Thank you very much for the suggestion. The sentence was changed.
“For the study aims it is important to define the term of red meat and processed meat.”
COMMENT#7
L70 add another aim toward …. consequences from this consumption and possible future interventions. This could strengthen and improve the article.
RESPONSE#7
As it was mentioned in the RESPONSE#1, the purpose of the study was to evaluate consumption using the budget method and to assess the frequency of consumption on the basis of the food propensity questionnaire method. The study did not investigate the effect of red and processed meat consumption on health.
COMMENT#8
Make sure you clearly state that the study presented in this article is based on analysis of a secondary data from Statistics Poland from the start. I noted it is present in L111. Also, is this data publicly available? L133”The interviewers tried to contact 4249 people” Please
RESPONSE#8
Only data on the amount of meat consumption from the household budget survey was based on analysis of a secondary data from Statistics Poland. Unpublished data of Statistics Poland purchased by the Institute was used to carry out this analysis. Data from Nationwide Dietary Survey in Poland was collected according to the European Food Safety Authority (EFSA) guidance on the EU Menu methodology by authors of this publication. These are raw data.
”The interviewers tried to contact 4249 people” seems necessary as it informs about the number of people whom the interviewers tried to contact and to whom the response rate relates.
COMMENT#9
L77 How the random sampling was done? L79-82 not so clear
COMMENT#10
L82 Which households? Perhaps Polish, which country areas, major cities.
RESPONSE#9-10
Thank you very much for the remark. The following text was added for clarification:
“The random sampling of the Polish households samples was based on the rules of the Statistics Poland. Firstly the sampling scheme covered the big cities (more than 100 thousands habitants) then the rest towns and villages. The first stage sampling frame was based on the records of statistical regions designed for the National Census including the dwelling specifications. The second stage sampling frame was based on the registers of dwellings in the selected primary sampling units. Each month, different households participated in the survey [27].”
COMMENT#11
How the Household budget survey was distributed? Is this part of the Statistics Poland national survey or your study own? It is not quite clear. L88-89 How the survey established knowledge of the following living conditions: “food consumption, housing conditions and equipment of the households with durables”.
COMMENT#12
L89-91 needs clarity.
RESPONSE#11-12
Thank you very much for the remark. The text was changed for clarification:
“Household budget survey which is yearly provided by Statistics Poland gives an information on the living conditions of the population. In the respect of food it provides the information about the consumed amount expressed per one person per household.. Each participated household received a special diary for a month, where registered the food quantities purchased, obtained free or derived from individual farm, garden or business activity [27].”
COMMENT#13
If there is no concrete data of the food quantities in catering establishments L93-94, please give references to the usual quantities, what is the average serve?
COMMENT#14
L96 Why it is internationally underestimated?
RESPONSE#13-14
Thank you very much for the remarks.
The household budget survey does not include the data of food consumed out of home.
We shorted the text to be more precise.
“The methodology used does not include food quantities consumed in catering establishments, canteens, hospitals, nurseries, kindergartens, etc. [27].”
COMMENT#15
L137- 138 “From among adults, persons who declared in the survey that they consumed meat were 137 selected (n=1,651, 812 men, 839 women).” Please rewrite for clarity.
RESPONSE#15
This information has been removed from the Materials and Methods as it is part of the Results. The information on the number and percentage of people who consumed meat provided in the Results has been refined with the data on the number of men and women.
The text was changed.
“Ninety percent of the 1,831 subjects (89% of 812 men and 91% of 839 women) declared that they consumed meat and meat products during the 12 months prior to the study.”
COMMENT#16
L155-164 Polish Academy of Science survey, household budget survey, Statistics Poland, PANEU etc.? It’s a bit confusing on which exactly study data collections are based this article. Please make it clear.
RESPONSE#16
This section refers only to food propensity questionnaire survey. But the adequate clarification was added:
“The food propensity questionnaire (FPQ) as the part of the nationwide dietary cross-sectional survey was used to collect data of frequency of consumption of selected meat and meat products.”
COMMENT#17
Before the concrete results by year are presented is needed a sentence or two explaining the overall results across years like meat consumption of both types processed and unprocessed meat varied between years…… and etc. L183-186 Why do you think such observation was noted “unprocessed red meat was eaten in rural than in urban area”? Same with red meat L187-190. Are there any other events that predisposed such results? Why do you think these products were consumed in the household of retirees and pensioners (L190, L204 in relation to chicken)? What the data reveals? Are there any explanations for this trend? Some is explained L391-400 which is great.
RESPONSE#17
We tried to explain the variation in consumption data from household budget survey according to place of residence and type of household in the discussion.
Taking into account your suggestion, we cited the results of German survey showing the diversification of meat consumption depending on the place of residence:
“In some countries, meat consumption also varies depending on the place of residence. Results of the German National Nutrition Survey II from 2005-2007 indicated that high meat consumption was more prevalent among persons living in small cities and rural areas in comparison with large cities [24].”
COMMENT#18
L201 Why the poultry consumption has not changed in the last decade?
RESPONSE#18
We would like to inform that in the last decade there were some changes in dietary habits throw reducing red meat consumption in accordance with actual recommendations. Additional explanation was provided in the discussion (RESPONSE#22).
COMMENT#19
Table 1 needs to be presented also in a graph for the trend to be seen.
RESPONSE#19
The additional figure was added into the result section.
COMMENT#20
L300 These references are great but how these are applicable to the results of the presented study? Please make your case, not just repeating the data shown in the survey results L323-325..
RESPONSE#20
Thank you for the remark, however it is not a repetition, but a conversion to g/week so that the recommendations can be referred to in the discussion section.
COMMENT#21
L332-333 What this leads to? Why it is good or bad?
RESPONSE#21
Thank you very much for the remark. The text was changed:
“The consumption of the processed meat in the amount 1.3 times over 50 g/day could increase the risk of non-communicable diseases in Poland.”
COMMENT#22
L340 Why it is a decline in red meat? Any particular reasons? Some trends? Price increase, availability problems etc.?
RESPONSE#22
Thank you very much for the remark.
We have added an explanation to the discussion (also for COMMENT#18):
“This could be due to the fact that in the years 2010-2020 the prices of meat increased in Poland, but poultry prices to a lesser extent than pork and beef prices [40,41,42]. Following a slight increase in processed meat consumption between 2000 and 2010, there was a decline in the next decade, especially for red meat which could be resulted by the growth of the price and possibly from a growing interest in healthy eating. In 2009, Poland developed revised dietary guidelines, which recommended to eat meat in moderation, including limiting the consumption of red meat [43].”
COMMENT#23
L377-378 -Why poultry was more often consumed than red meat.
RESPONSE#23
Thank you very much for the remark. The text was added:
“It could be also explained as the result of price growth of red meat and changing the eating habits.”
COMMENT#24
L386 “In Poland, we observed a different situation: the consumption of processed meat 386 exceeded the consumption of unprocessed red meat. It should be noted that the data 387 from the household budget survey refer to raw, not to cooked meat.” Is a very interesting finding. Why do you think is that?
RESPONSE#24
Thank you very much for the remark. The following explanation has been added:
“Poles often eat sandwiches, especially for breakfastA frequent addition to sandwiches are cold cuts and sausages. This can cause a higher consumption of processed meat” [39,50].
COMMENT#25
Combine the single sentence paragraphs (starting L40 and L44) and L197-204; L209-212; L301; L389; L432 etc. as usually there are 3 to 8 sentences in a paragraph. Please check the whole text as there are some tendency.
RESPONSE#25
We analysed the manuscript in this regard and some paragraphs were combined.
COMMENT#26
L402 Are there any explanations for the “differences for cold cuts, which were eaten more often in rural than in urban areas”?
RESPONSE#26
Unfortunately, the authors of the cited study do not explain the reasons for this difference. They also did not provide information whether the groups of respondents from the urban and the rural areas differed from each other in terms of such socio-demographic characteristics as the educational level or the economic status, which could have an impact on it. Due to the insufficient amount of information on the study groups and the lack of conclusions in this regard by the authors of this article, we did not comment on this difference in our publication.
The number of reference was replaced:
“In the case of meat, there were significant differences for cold cuts, which were eaten more often in rural than in urban areas [48].”
COMMENT#27
Please give arguments for presenting the studies in the US, Canada, British and Mexico L417-419 and L432. Is this in comparison with your Polish study? It is brought a bit abrupt and without any linking explanation.
RESPONSE#27
We want to show whether a similar relationship has also been observed in nationally-representative cross-sectional survey from other countries in Europe and in the North America, where the dietary recommendations for meat consumption are similar. Due to the fact that the European data are more comparable with the Polish ones, we changed the order of the compared studies in the text: first European (data from 10 European countries), then American. We have also detailed information about British and American studies. Additionally we cited German study.
The text was changed:
“The review of data from ten European countries showed that mean intakes of total meat across Europe were higher in men (84–218 g/day) than in women (64–163 g/day) [47]. The aforementioned national survey in Germany indicated much higher meat consumption by men (155 g/day) than by women (87 g/day) [24]. However ac-cording to UK data from the National Diet and Nutrition Survey rolling programme (2008–09 to 2018–19) there was no significant difference in meat intake between men and women from the UK when the intake was expressed as a percentage of food energy [45]. The relationship between sex and meat consumption was also assessed by authors of nationally-representative cross-sectional survey comparison of dietary recalls from Canada, Mexico, and the United States. In US and Canada unprocessed red and processed meat intake was higher for men than women among consumers. No such relationship has been found in Mexico [49].”
COMMENT#28
Full stop in missing after “sources” L92
RESPONSE#28
The sentence was removed.
COMMENT#29
L238 The subheadings 3.2.2. and 3.2.4. can’t be distinguished from the text. Make it italic or bold depending on the journal’s requirements.
RESPONSE#29
According to the rules the subsubsection must be in normal text.
COMMENT#30
L281 Full stop is not needed at the end of the subheading.
RESPONSE#30
The change was done.
COMMENT#31
Conclusion must be a bit more conclusive as the moment seems like it is presenting a summary in bullet points without actually including them which is not necessary. Although the major points were summarized, the conclusion did not fully utilize the chance the authors to have their last word on the subject. Try to make it more cohesive.
COMMENT#32
The authors should be complimented on taking on this important area of research
RESPONSE#31-32
Thank you for the remark. The Conclusions has been corrected.
Although in recent years, in Poland there has been noted decline in the consumption of red meat, both unprocessed and processed, the consumption of this type of food is still high, especially the amount of processed meat in the diet exceeds the recommendations. High consumption of processed meat may increase the risk of chronic diseases. More meat and processed products were eaten in rural than in urban area. The highest consumption was noted in the households of retirees and pensioners. Among the different types of meat poultry was often consumed. It seems positive because it is recommended to choose this type of meat instead of red meat. Unfortunately the frequent consumption of pork, especially among men, and processed meat (cold cuts, such as: ham, loin and in men sausages and bacon) was also observed. Depending on the type of meat and meat products the relationships were observed between the frequency of their consumption and socio-demographic characteristics, such as sex, age, education level, economic status and health status. In Poland it is very important to follow up activities aimed at reducing the amount and frequency of consumption of mainly various types of processed meat, as well as unprocessed red meat. There is a need to educate the public in the field of healthy eating, including the health effects of consuming too much red meat and processed meats, as well as the sustainability development and environmental impact of too large meat production. Educational activities should be directed in particular to the population groups most exposed to negative health effects caused by excessive consumption of red and processed meat..
The authors would like to express their sincere thanks for the efforts in reviewing the manuscript and for all their comments and suggestions towards improving the proposed article.
Reviewer 2 Report
The paper assessed the quantity and frequency of meat consumption in Poland. Two datasets were used: a household budget survey from 2000, 2010, 2020, and a survey employing a food propensity questionnaire in 1831 adults in 2019-2020. The study found frequent red and processed meat consumption in Poland, with consumption of processed meat exceeding public health recommendations, thereby increasing the risk of chronic diseases.
The topic is highly relevant for various disciplines and has numerous practical implications. Overall, the study is well-designed, and the paper is well-written. There are several suggestions I would propose to the authors:
1.
“It can also provide the body with unfavourable saturated fatty acids or cholesterol. In 26 addition, processed meat often contains high amounts of sodium.”
I suggest adding a reference, as it is unclear whether the previous one also refers to these sentences.
2.
“It is assumed that the unfavourable effect of red and processed meat results from the fat 40 content, including saturated fatty acids, heam iron, heat-includec carcinogens e.g. het- 41 erocyclic aromatic amines, and in the case of processed meat also from the addition of 42 salt and other preservatives [3,12]. 43 Nevertheless, many experts recommend to reduce red and processed meat consump- 44 tion.”
“Nevertheless” is misused here, as the previous sentences show unfavourable effect of meat consumption, followed by an expert recommending meat reduction.
3.”
“Moreover 53 livestock farming also accounts for a significant proportion of water consumption, which 54is largely used to irrigate feed crops [17]. 55”
I suggest adding a few sentences with references regarding other negative impacts of livestock farming (reducing biodiversity, etc.).
4.
“The aim of the study was to assess the current consumption of meat, especially red and 66 processed meat in Poland and the trends in this regard in the last 2 decades on the basis 67 of household budget survey. The aim of this study was also to analyse the frequency of 68 selected meat and meat products consumption associated with sociodemographic fac- 69 tors in a representative sample of adult inhabitants of Poland. 70”
One of the aims of the study was to examine sociodemographic factors, yet (at least) a short literature review is missing in the Introduction section.
While in the Discussion section, there are references for the U.S., Canada and Latin America (lines 447-450), references are needed (see, for example, below) that refer to European countries, including Eastern-European, which makes a comparison (i.e. contextualization) more meaningful. Specifically, the authors may take advantage of MDPI journals and other publications on determinants of meat consumption and include them for an interested reader in the Introduction section, for example, examining sex, education, income, age, size of residential settlement, etc. See, for example:
a)
https://doi.org/10.3390/su132313036
b)
Gossard, M.H.; York, R. Social Structural Influences on Meat Consumption.Hum. Ecol. Rev.2003,10, 1–9.
c)
Koch, F.; Heuer, T.; Krems, C.; Claupein, E. Meat consumers and non-meat consumers in Germany: A characterisation based on rresults of the German National Nutrition Survey II.J. Nutr. Sci.2019,8, e21
d)
Einhorn, L. Meat consumption, classed? Osterr. Z. für Soziologie 2021, 46, 125–146.
e)
Vandermoere, F.; Geerts, R.; de Backer, C.; Erreygers, S.; van Doorslaer, E. Meat Consumption and Vegaphobia: An Exploration of the Characteristics of Meat Eaters, Vegaphobes, and Their Social Environment. Sustainability 2019, 11, 3936.
5.
»sources The 92«
A full stop is missing.
“These feature is common 94”
“This” (or “features are”)
6.
“2.1.3. Data on meat consumption 97 For the purposes of this study, the data collected by Statistics Poland were trans- 98 formed to evaluate the consumption of various types of meat. The consumption of raw 99 (unprocessed) and processed meat was calculated. Raw meat included: total red meat 100”
I suggest adding items/questions on meat consumption in grams to Supplementary material for an interested reader.
7.
“Dietary recommendations often relate to the consumption of meat, including red 301 and processed meat.”
I suggest single-sentence paragraphs to be joined in a meaningful matter. In addition, reference is welcome after above-stated sentence.
8.
“Fish, eggs, legumes, mycopro- 307 tein-based foods and seitan are mentioned as meat substitutes [2, 14]. ”
Fish, eggs, dairy and poultry, are often mentioned in the article as a healthier choice than meat; however, a reader might get an impression that the authors see them as the healthiest choice. I suggest the authors note several studies that indicate the healthiest meat, dairy, eggs and fish alternative food choices, such as legumes, soy, … (e.g., ).
9.
“In the Polish FBDG, special attention is paid to limiting the consumption of red and 309 processed meat to 500 g per week [26]. It is recommended not to eat meat for at least 1 310 day a week, to choose poultry and to replace meat with fish, eggs and plant products 311 such as legumes and nuts.” XYZ
Putting FBDG in context would be helpful to a reader, for example, by referencing studies on the health effect of poultry and fish in replacement studies, e.g.,: XYZ
10.
“Data from the household budget survey presented in this manuscript indicate that 323”
I suggest replacing “manuscript” with “study”.
11.
“Taking 330 into account the EFSA conclusions, consumption of unprocessed red meat has not r e- 331 cently increased the risk of chronic disease. On the other hand, the consumption of pr o- 332 cessed meat by about 1.3 times exceeded the amount estimated as safe.”
It is unclear what “recently” refers to in this sentence.
In addition, does EFSA say that red meat does not increase chronic disease risk, despite WCRF/AICR, etc.?
12.
“The review of data from ten European countries showed that mean intakes of total 420 meat across Europe were higher in men (84–218 g/day) than in women (64–163 g/day) 421 [30]. 422”
I suggest combining it into a single paragraph.
13.
“In British studies, the older adults consumed less meat expressed as a percentage of 432 food energy compared with younger adults [28]. 433
In Mexico, the consumption of total meat in the elderly was lower than in the 434 younger age [32]. Also in Canada and the US, elderly people consumed less meat, how- 435 ever the differences were not meaningful. 436”
I suggest combining it into a single paragraph.
14.
“The authors of this paper argue that GDP growth to some 461 extent leads to an increase in meat consumption, however, with a GDP of around USD 462 40,000 per capita, the increase in economic well-being in the country does not lead to 463 growth in meat consumption. 464”
The authors argue that on what basis? A reference would be welcome.
15.
The Conclusion should be written in a single paragraph.
Despite these comments, the paper is clearly written, relevant to the field and presented in a well-structured manner. I suggest the authors take these comments into account in the revised version of the paper, which I would be very happy to read, and I congratulate the authors on the study well done.
Author Response
AUTHORS’ RESPONSE TO THE COMMENTS FROM THE REVIEWER#2
Thank you for all the comments. We have carefully analysed all valuable comments and suggestions that allow the Authors to increase the scientific soundness of our paper. The authors modified the manuscript to improve the quality of the paper. Our responses are listed below.
COMMENT#1
“It can also provide the body with unfavourable saturated fatty acids or cholesterol. In 26 addition, processed meat often contains high amounts of sodium.”
I suggest adding a reference, as it is unclear whether the previous one also refers to these sentences.
RESPONSE#1
Thank you very much for the remark. The cited references were replaced to the end of the sentence.
COMMENT#2
“It is assumed that the unfavourable effect of red and processed meat results from the fat 40 content, including saturated fatty acids, heam iron, heat-includec carcinogens e.g. het- 41 erocyclic aromatic amines, and in the case of processed meat also from the addition of 42 salt and other preservatives [3,12]. 43 Nevertheless, many experts recommend to reduce red and processed meat consump- 44 tion.”
“Nevertheless” is misused here, as the previous sentences show unfavourable effect of meat consumption, followed by an expert recommending meat reduction.
RESPONSE#2
Thank you for the remark. The sentence was changed.
“Many experts recommend to reduce red and processed meat consumption.”
COMMENT#3
“Moreover 53 livestock farming also accounts for a significant proportion of water consumption, which 54is largely used to irrigate feed crops [17]. 55”
I suggest adding a few sentences with references regarding other negative impacts of livestock farming (reducing biodiversity, etc.).
RESPONSE#3
Thank you very much for the suggestion. The additional reference was added.
“About one-third of the world’s water consumption is for producing animal products. Animal agriculture is responsible for 20 to 33 percent of all fresh water consumption in the world [21,22]. Daily water requirements of pigs depending on age is ranged between 3–12L/day and for cattle is ranged between 25-160 l/day [23]”.
COMMENT#4
“The aim of the study was to assess the current consumption of meat, especially red and 66 processed meat in Poland and the trends in this regard in the last 2 decades on the basis 67 of household budget survey. The aim of this study was also to analyse the frequency of 68 selected meat and meat products consumption associated with sociodemographic fac- 69 tors in a representative sample of adult inhabitants of Poland. 70”
One of the aims of the study was to examine sociodemographic factors, yet (at least) a short literature review is missing in the Introduction section.
While in the Discussion section, there are references for the U.S., Canada and Latin America (lines 447-450), references are needed (see, for example, below) that refer to European countries, including Eastern-European, which makes a comparison (i.e. contextualization) more meaningful. Specifically, the authors may take advantage of MDPI journals and other publications on determinants of meat consumption and include them for an interested reader in the Introduction section, for example, examining sex, education, income, age, size of residential settlement, etc. See, for example:
a)
https://doi.org/10.3390/su132313036 (Kirbis A)
b)
Gossard, M.H.; York, R. Social Structural Influences on Meat Consumption.Hum. Ecol. Rev.2003,10, 1–9.
c)
Koch, F.; Heuer, T.; Krems, C.; Claupein, E. Meat consumers and non-meat consumers in Germany: A characterisation based on rresults of the German National Nutrition Survey II.J. Nutr. Sci.2019,8, e21
d)
Einhorn, L. Meat consumption, classed? Osterr. Z. für Soziologie 2021, 46, 125–146.
e)
Vandermoere, F.; Geerts, R.; de Backer, C.; Erreygers, S.; van Doorslaer, E. Meat Consumption and Vegaphobia: An Exploration of the Characteristics of Meat Eaters, Vegaphobes, and Their Social Environment. Sustainability 2019, 11, 3936.
RESPONSE#4
Thank you for the remark. The suggested references were included into the introduction and discussion sections.
“The consumption of meat is determined by sociodemographic factors. Firstly the important meaning has the sex of a consumer. Women much more reduce meat consumption than men. Besides The locality has influence of meat consumption as well. Small cities and villages citizens choice is to eat more often meat than large cities habitants [24]. Consumers with high education level and high income could easier adopt their dietary habits towards meat reducing or meat excluding in their diets [25]. However the income influence on meat consumption choice is not obvious [26].”
“In some countries, meat consumption also varies depending on the place of residence. Results of the German National Nutrition Survey II from 2005-2007 indicated that high meat consumption was more prevalent among persons living in small cities and rural areas in comparison with large cities [24].”
“The aforementioned national survey in Germany indicated much higher meat consumption by men (155 g/day) than by women (87 g/day) [24].”
“Trend of reducing meat consumption with increasing age was observed in Germany [24].”
“In other countries, the level of education also influenced meat consumption. Slovenian study showed that the education level had impact to the frequency of meat consumption and sustainable attitudes; higher educational level was an independent predictor of lower meat consumption [53]. In Germany consumers who had characterized by the high education level more often reduced meat consumption than people with low education level [24].”
“The impact of socio-demographic factors on food choices is quite complex. A consumer study by Einhorn found that younger and well-educated respondents primarily perceive their diet as something that can be experimented with. Mostly highly educated and financially well-off people want to try new foods and have advanced knowledge of nutrition. When time or money is scarce, respondents show little interest in getting involved in food planning, purchasing, and preparation [25].”
COMMENT#5
»sources The 92«
A full stop is missing.
“These feature is common 94”
“This” (or “features are”)
RESPONSE#5
Thank you very much for the remark. The sentences were changed.
“Each participated household received a special diary for a month, where registered the food quantities purchased, obtained free or derived from individual farm, garden or business activity [27]. The methodology used does not include however food quantities consumed in catering establishments, canteens, hospitals, nurseries, kindergartens, etc. [27].”
COMMENT#6
“2.1.3. Data on meat consumption 97 For the purposes of this study, the data collected by Statistics Poland were trans- 98 formed to evaluate the consumption of various types of meat. The consumption of raw 99 (unprocessed) and processed meat was calculated. Raw meat included: total red meat 100”
I suggest adding items/questions on meat consumption in grams to Supplementary material for an interested reader.
RESPONSE#6
Thank you very much for the suggestion, however this part of study came from the household budget survey and it is not possible to receive any questionnaire.
The more clarified information was added.
“The random sampling of the Polish households samples was based on the rules of the Statistics Poland. Firstly the sampling scheme covered the big cities (more than 100 thousands habitants) then the rest towns and villages. The first stage sampling frame was based on the records of statistical regions designed for the National Census including the dwelling specifications. The second stage sampling frame was based on the registers of dwellings in the selected primary sampling units. Each month, different households participated in the survey [27].”
COMMENT#7
“Dietary recommendations often relate to the consumption of meat, including red 301 and processed meat.”
I suggest single-sentence paragraphs to be joined in a meaningful matter. In addition, reference is welcome after above-stated sentence.
RESPONSE#7
Thank you very much for the suggestion. The paragraphs were joined.
“Dietary recommendations often relate to the consumption of meat, including red and processed meat. Food-based dietary guidelines (FBDGs) of most European countries recommend limiting meat consumption to around 300–600 g per week and not eating meat every day…..”
COMMENT#8
“Fish, eggs, legumes, mycopro- 307 tein-based foods and seitan are mentioned as meat substitutes [2, 14]. ”
Fish, eggs, dairy and poultry, are often mentioned in the article as a healthier choice than meat; however, a reader might get an impression that the authors see them as the healthiest choice. I suggest the authors note several studies that indicate the healthiest meat, dairy, eggs and fish alternative food choices, such as legumes, soy, … (e.g., ).
RESPONSE#8
Thank you for your suggestion. These products are indicated in the recommendations of many countries as a healthier alternative of meat. The sentence was changed.
“In recommendations of many countries fish, eggs, legumes, mycoprotein-based foods and seitan are mentioned as meat substitutes [2,14].”
COMMENT#9
“In the Polish FBDG, special attention is paid to limiting the consumption of red and 309 processed meat to 500 g per week [26]. It is recommended not to eat meat for at least 1 310 day a week, to choose poultry and to replace meat with fish, eggs and plant products 311 such as legumes and nuts.” XYZ
Putting FBDG in context would be helpful to a reader, for example, by referencing studies on the health effect of poultry and fish in replacement studies, e.g.,: XYZ
RESPONSE#9
Thank you for the remark. The adequate reference was added.
“Meta-analysis by Guasch-Ferré et al. indicates that replacing red meat with high-quality plant protein sources, but not with fish or low-quality carbohydrates, provides to more favorable changes in blood lipids and lipoproteins.”
COMMENT#10
“Data from the household budget survey presented in this manuscript indicate that 323”
I suggest replacing “manuscript” with “study”.
RESPONSE#10
Thank you for the remark. The change in the sentence was made.
“Data from the household budget survey presented in this study indicate that the consumption of unprocessed red meat in Poland in 2020 was 311 g/person/week, whilst the consumption of processed meat, including red – 451 g/person/week.”
COMMENT#11
“Taking 330 into account the EFSA conclusions, consumption of unprocessed red meat has not r e- 331 cently increased the risk of chronic disease. On the other hand, the consumption of pr o- 332 cessed meat by about 1.3 times exceeded the amount estimated as safe.”
It is unclear what “recently” refers to in this sentence.
In addition, does EFSA say that red meat does not increase chronic disease risk, despite WCRF/AICR, etc.?
“Plausible mechanisms through which consumption of processed meat, and to a lesser extent of unprocessed red meat, could increase the risk of CVD, T2DM and certain types of cancer, include the intake of high amounts of sodium and other preservatives (for processed meat only), haem iron and heat-induced carcinogens (process contaminants), as well as the unfavourable fatty acid profile (Al-Shaar et al., 2020; Papier et al., 2021)……. Most FBDGs recommend limiting meat intake, some suggesting specifically the reduction of unprocessed red and processed meat consumption”
RESPONSE#11
Thank you very much for the remark. The clearing of the sentence was improved.
“Taking into account the EFSA conclusions, consumption of unprocessed red meat has not increased the risk of chronic disease [2]. The consumption of the processed meat in the amount 1.3 times over 50 g/day could increase the risk of non-communicable diseases in Poland.”
COMMENT#12
“The review of data from ten European countries showed that mean intakes of total 420 meat across Europe were higher in men (84–218 g/day) than in women (64–163 g/day) 421 [30]. 422”
I suggest combining it into a single paragraph.
RESPONSE#12
Thank you very much for the suggestion. The paragraphs were joined.
“The review of data from nationally representative nutrition surveys from 2003-2014 in ten European countries showed that meat consumption varied across countries [47]. Total meat consumption in adults ranged from 75-93 g/day in Sweden to 191-211 g/day in Finland.”
COMMENT#13
“In British studies, the older adults consumed less meat expressed as a percentage of 432 food energy compared with younger adults [28]. 433
In Mexico, the consumption of total meat in the elderly was lower than in the 434 younger age [32]. Also in Canada and the US, elderly people consumed less meat, how- 435 ever the differences were not meaningful. 436”
I suggest combining it into a single paragraph.
RESPONSE#13
Thank you very much for the suggestion. The paragraphs were joined.
“In British studies, the older adults consumed less meat expressed as a percentage of food energy compared with younger adults [45]. In Mexico, the consumption of total meat in the elderly was lower than in the younger age. Also in Canada and the US, elderly people consumed less meat, however the differences were not meaningful [49].”
COMMENT#14
“The authors of this paper argue that GDP growth to some 461 extent leads to an increase in meat consumption, however, with a GDP of around USD 462 40,000 per capita, the increase in economic well-being in the country does not lead to 463 growth in meat consumption. 464”
The authors argue that on what basis? A reference would be welcome.
RESPONSE#14
Thank you very much for the remark. This is the commentary from the reference. In order to clarify the sentence, the reference number was replaced to the end of the paragraph.
“The authors of this paper argue that GDP growth to some extent leads to an increase in meat consumption, however, with a GDP of around USD 40,000 per capita, the increase in economic well-being in the country does not lead to growth in meat consumption [44].”
COMMENT#15
The Conclusion should be written in a single paragraph.
COMMENT#16
Despite these comments, the paper is clearly written, relevant to the field and presented in a well-structured manner. I suggest the authors take these comments into account in the revised version of the paper, which I would be very happy to read, and I congratulate the authors on the study well done.
RESPONSE#15-16
Thank you for the remark. The Conclusions has been corrected.
Thank you for the overall evaluation of the paper.
Although in recent years, in Poland there has been noted decline in the consumption of red meat, both unprocessed and processed, the consumption of this type of food is still high, especially the amount of processed meat in the diet exceeds the recommendations. High consumption of processed meat may increase the risk of chronic diseases. More meat and processed products were eaten in rural than in urban area. The highest consumption was noted in the households of retirees and pensioners. Among the different types of meat poultry was often consumed. It seems positive because it is recommended to choose this type of meat instead of red meat. Unfortunately the frequent consumption of pork, especially among men, and processed meat (cold cuts, such as: ham, loin and in men sausages and bacon) was also observed. Depending on the type of meat and meat products the relationships were observed between the frequency of their consumption and socio-demographic characteristics, such as sex, age, education level, economic status and health status. In Poland it is very important to follow up activities aimed at reducing the amount and frequency of consumption of mainly various types of processed meat, as well as unprocessed red meat. There is a need to educate the public in the field of healthy eating, including the health effects of consuming too much red meat and processed meats, as well as the sustainability development and environmental impact of too large meat pro-duction. Educational activities should be directed in particular to the population groups most exposed to negative health effects caused by excessive consumption of red and pro-cessed meat..
The authors would like to express their sincere thanks for the efforts in reviewing the manuscript and for all their comments and suggestions towards improving the proposed article.
Round 2
Reviewer 1 Report
Thank you for the thorough revision of your paper. It reads well now.
Reviewer 2 Report
The paper is much improved. I suggest English language editing.